

# Data Assimilation with An Improved Particle Filter and Its Application in TRIGRS Landslide Model

Changhu Xue[1], Guigen Nie[1], Haiyang Li[1], Jing Wang[1]

[1]GNSS Research Center, Wuhan University, Wuhan, 430079, China

*Correspondence to*: Guigen Nie(ggnie@whu.edu.cn)

**Abstract.** Particle filter has become a popular algorithm in data assimilation for its capability to handle non-linear or non-Gaussian state-space models, while it still be seriously influenced by its disadvantages. In this work, the particle filter algorithm is improved, proposed two methods to overcome the particle degeneration and improve particles' efficiency. In this algorithm particle-propagating and resample method are ameliorated. The new particle filter is applied to Lorenz-63 model, verified its feasibility and effectiveness using only 20 particles. The root mean square difference(RMSD) of estimations converge to stable when there are more than 20 particles. Finally, we choose a 10 * 10 grid slope model of TRIGRS and carry out an assimilation experiment. Results show that the estimations of states can effectively correct the running-offset of the model and the RMSD is convergent after 3 days assimilation.

**Key words.** Data assimilation, particle filter, nonlinear model, Lorenz-63, landslide model

## 1 Introduction

A mount of mountainous areas have suffered frequent landslide disasters all over the world. People living in mountainous areas are faced with the threat of landslide disasters. A destructive landslide occurred on June 24, 2017 located at 103°39' 03"E, 32°04' 09"N, Maoxian County, Sichuan province, China, caused a huge loss of personal and property. Works of landslide monitoring, analysis and forecasting are crucial. Many numerical modeling methods of slope evolution are proposed and developed recently such as discontinuous deformation analysis (DDA) (Shi 1992, Jing, Ma et al. 2001, Ma, Kaneko et al. 2011) and distinct elements methods (DEM) (Lorig and Hobbs 1990, Marcato, Fujisawa et al. 2007, Li, He et al. 2012). Iverson carried out the TRIGRS program to predict the stability of landslides in response to rainfall. It is a raster-based model, depends on time for transient rainfall infiltration(Iverson 2000, Baum 2008). Jiang adopted the Ensemble Kalman filter to landslide movement model in relation to hydrological factors, which introduce data assimilation (DA) to landslide.

Data assimilation is a common approach to solve an estimation of optimal state in dynamic systems. With DA algorithms and operators, DA merges different scales of observations into dynamic models to take advantage of all information. Many DA



algorithms have been developing and improving in recent years, in which particle filter (PF) is a popular algorithm for its availability under conditions of nonlinear and non-Gaussian distributed models (Arulampalam, Maskell et al. 2002, Moradkhani, Hsu et al. 2005). Increasing applications and improvements of PF have been researched recently in DA or other fields. Salamon, *et al.* (Salamon and Feyen 2009) applied the residual resampling particle filter (RRPF) to assess parameter,

precipitation, and predictive uncertainty in rainfall–runoff model. Thirel, *et al.* (Thirel, Salamon et al. 2013) assimilated the snow cover area in physical distributed hydrological models and MODIS satellite data to improve the pan-European flood forecasts. Mattern, *et al.*(Mattern, Dowd et al. 2013) carried out assimilation experiments for a three-dimensional biological ocean model and satellite observations and verified the feasibility of biological state estimation with sequential importance resampling (SIR) for realistic models.

However, large computational complexity and particle degradation or collapse are still obstacles in PF. To solve these problems, some resample algorithms have been proposed. One improvement is adding an item related to observations, to make the proposal density dependent on the future observations, accordingly most particles could situate into the range of observation error (van Leeuwen 2010). It can get good results to using only 10~20 particles in high dimensional assimilation experiments. But the number of key particles are reduced when the system variance is larger than the observed variance, and the values of

added items are uncertain. Another improvement is to replace the duplicated process by generating a Halton sequence in residual resampling(Zhang, Qin et al. 2013). The disordered particle sets are turned into ordered and too few particles can hardly describe the posterior PDF better.

In this paper, a new resampling approach is proposed to improve the above method, keeping both particles' diversity and efficiency. Applying to Lorenz-63 model using different numbers of particles range from 10~200, this method has shown its

efficiency and sensitivity to the number of particles. Finally, we choose a slope movement model with a 10*10 size grid, applying the assimilation algorithm and TRIGRS program to predict and improve the prediction of safety factors and deformations of the landslide.

## 2 Improvements of Residual Resampling Particle Filtering

In sequential importance sampling, the state vector is represented by a set of particles

$$x_k = f(x_{k-1}) + G_k(x_{k-1})\varepsilon_k \tag{1}$$

where $x$ is the state vector with initial PDF $p(x_0)$, $k$ is the subscript of time steps, $\varepsilon_{k-1}$ is system noise with zero mean at step $k$-1, $f(\cdot)$ is model operator. Initial $N$ particles are sampled from $p(x_0)$. The observation equation is

$$z_k = h(x_k) + \eta_k \tag{2}$$

where $z$ is the observation vector, $h(\cdot)$ is observation operator. Weights of particles are calculated by (3), and normalized to

get $w_k^i$





$$\tilde{w}_k^i = w_{k-1}^i \cdot \frac{p(z_k \mid x_k^i) p(x_k^i \mid x_{k-1}^i)}{q(x_k^i \mid x_{k-1}^i, z_k)} \qquad (3)$$

$$w_k^i = \frac{\tilde{w}_k^i}{\sum\limits_{j=1}^{N} \tilde{w}_k^j} \qquad (4)$$

where $p(z_k \mid x_k^i)$ is the likelihood of observation, $q(x_k^i \mid x_{k-1}^i, z_k)$ is the proposal function.

Residual resample is a way to solve the problem of particle degeneracy which is an unavoidable trouble in PF. To keep most

particles effective, low-weight particles are removed and high-weight particles are duplicated. But with the recursive progress the particle sets can hardly represent the prior PDF due to the declining of particles diversity.

Some improvements about residual resample algorithm are proposed in this paper. Firstly, in the process of particle transferring, we choose

$$x_k^i = f(x_{k-1}^i) + \hat{\varepsilon}_{k-1} + J_k[z_k - h(\hat{x}_{k-1})] \qquad (5)$$

where $J_k$ is a coefficient like the "gain" in extended Kalman filter:

$$\left. \begin{array}{l} J_k = D_{k/k-1} B_k^{\mathrm{T}} [B_k D_{k/k-1} B_k^{\mathrm{T}} + R_k]^{-1} \\ D_{k/k-1} = A_{k-1} D_{k-1/k-1} A_{k-1}^{\mathrm{T}} + G_{k-1}(\hat{x}_{k-1}) Q_{k-1} G_{k-1}^{\mathrm{T}}(\hat{x}_{k-1}) \end{array} \right\} \qquad (6)$$

in which $A_k$, $B_k$ are the linearization parameter of $f(\bullet)$ and $h(\bullet)$, respectively:

$$A_k = \frac{\partial f_k}{\partial x_k}(\hat{x}_k), \quad B_k = \frac{\partial f_k}{\partial x_k}(\hat{x}_{k/k-1}) \qquad (7)$$

$D_{k/k}$ is estimation variance of state $x_k$ at step $k$. This process is equal to translate particles close to observations. But the value

of $J_k$ is hard to determine because the variance of state estimation $D_{k-1/k-1}$ in PF is difficult to compute. To simplify the calculation, suppose that the translated particles are a series of virtual observations about the state at step $k$. Write the particle set as:

$$X_{k/k}^N = \left\{ x_{k/k}^i \right\}_{i=1,2,\dots,N} \qquad (8)$$

and replace $D_{k-1/k-1}$ with the variance of particles. To keep the value of $D_{k-1/k-1}$ unchanged before and after translation, we

choose the posterior particles at step $k$-1:

$$D_{k-1/k-1} = \mathrm{var}(X_{k-1/k-1}) \qquad (9)$$

Secondly, using the method of Zhang *et al.* (Zhang, Qin et al. 2013)to compute accumulative copy times (ACT), each parent particles with high weights regenerates a set of new particles. Differently, instead of duplicating or generating Halton sequence, it generates a series of normal-distributed particles:

$$\left\{ x_k^1, x_k^2, \dots, x_k^{ACT_i} \right\} \sim N\left( x_k^i, G_k\left( x_k^i \right) \right)$$



where $ACT_i$ is the ACT of the $i$th particle, the mean and variance are related on the value of parent. Accordingly, the resampled particle set is composed of some different particle sets which obey normal distribution. Assume that the $j$th progeny particle of $x_k^i$ is written as $x_k^{ij}$, the formula (3) can be written as:

$$\tilde{w}_k^{ij} = w_{k-1}^i \cdot \frac{p(z_k \mid x_k^{ij}) p(x_k^{ij} \mid x_{k-1}^i)}{q(x_k^{ij} \mid x_{k-1}^i, z_k)} \tag{10}$$

Shortly, the improved RRPF in this section can be implemented by the following steps:

Step 1: Draw initial particles $\{x_0^i\}$ from prior PDF $p(x_0)$.

Step 2: Compute the mean and variance of posterior particles at step $k$-1:

$$\bar{x}_{k\text{-}1/k-1} = \frac{1}{N} \sum_{i=1}^{N} x_{k\text{-}1/k-1}^i \tag{11}$$

$$D_{k\text{-}1/k-1} = \frac{1}{N-1} \sum_{i=1}^{N} (x_{k\text{-}1/k-1}^i - \bar{x}_{k\text{-}1/k-1})(x_{k\text{-}1/k-1}^i - \bar{x}_{k/k-1})^{\mathrm{T}} \tag{12}$$

Step 3: Using the new method in this section, compute the "gains" of particles:

$$D_{k/k-1} = \left[ \frac{\partial f_k}{\partial x_k}(\hat{x}_k) \right] D_{k\text{-}1/k-1} \left[ \frac{\partial f_k}{\partial x_k}(\hat{x}_k) \right]^{\mathrm{T}} + G_{k-1}(\hat{x}_{k-1}) Q_{k-1} G_{k-1}^{\mathrm{T}}(\hat{x}_{k-1}) \tag{13}$$

$$J_k = D_{k/k-1} \left[ \frac{\partial f_k}{\partial x_k}(\hat{x}_{k/k-1}) \right] \left\{ \left[ \frac{\partial f_k}{\partial x_k}(\hat{x}_{k/k-1}) \right] D_{k/k-1} \left[ \frac{\partial f_k}{\partial x_k}(\hat{x}_{k/k-1}) \right]^{\mathrm{T}} + R_k \right\}^{-1} \tag{14}$$

Step 4: Transfer the particles close to the observation:

$$x_k^i = f(x_{k-1}^i) + \hat{\varepsilon}_{k-1} + J_k[z_k - h(\hat{x}_{k-1})] \tag{15}$$

Step 5: Residual resampling. Each particle generates a set of normal-distributed progeny particles, and all progeny sets make up the resampled particle set:

$$\left\{ x_k^{i1}, x_k^{i2}, \dots, x_k^{iACT_i} \right\} = X_k^{iACT_i} \sim N\left( x_k^i, G_k\left( x_k^i \right) \right) \tag{16}$$

$$\left\{ X_k^{1ACT_1}, X_k^{2ACT_2}, \dots, X_k^{N\,ACT_N} \right\} = \left\{ x_k^{*i} \right\}_{i=1,2,\dots,N} \tag{17}$$

When $ACT_i = 0$, $X_k^{iACT_i}$ is empty set.

Step 6: Compute and normalize weights:

$$\tilde{w}_k^i = w_{k-1}^i \cdot p(z_k \mid x_k^i) \tag{18}$$

$$w_k^i = \frac{\tilde{w}_k^i}{\sum_{j=1}^{N} \tilde{w}_k^j} \tag{19}$$





Step 7: Compute the state estimation:

$$\hat{x}_{k/k} = \sum_{i=1}^{N} x_k^{*i} \cdot w_k^i \tag{20}$$

## 3 Application to Lorenz-63 model

We choose the Lorenz-63 model as an example to test the improved algorithm(Baines 2008). Parameters are given by:
$dt = 0.01$, $\sigma = 10$, $\rho = 28$, $\beta = 8/3$, the observation error $\sigma_{obs} = \sqrt{2}$, model transmission error based on time interval

$\sigma_{mod} = 2\sqrt{\Delta t}$. Initialize the filter with the starting point which is set to $(x_0, y_0, z_0) = (1.50887, -1.531271, 25.46091)$. The truth is obtained by the formula of the model recursively. Observations are generated form the truth by adding a disturbance every 40 time steps. Recurs 1000 steps, assimilate the observation with the model when observation exists at current step and recurs to next step when there is no observation.

Figure 1 shows the results of *x*-component using new PF with 20 particles. Note that the new PF procedure is closed to the truth with much fewer particles which is more efficient than standard PF with hundreds of particles. Compute the confidence interval with 95% level using the posterior particles every step. Figure 2 shows that the intervals contain observations at almost all the steps with observations exist. That means particle sets after translation are very closed to observations and true states. The evolution of all particles is displayed in figure 3, in which most particles are very closed to observations except for several
ones at moments with state changed obviously.  Consider the root mean square difference (RMSD) of the estimation with respect to particle numbers as the following formula

$$RMSD = \sqrt{\frac{1}{T}\sum_{t=1}^{T}(\hat{X}_t - X_t^{obs})^2} \tag{19}$$

where *T* is the period of assimilation, $\hat{X}_t$ and $X_t^{obs}$ are the assimilated value and the observation of state at time *t*. The

RMSE sequence is shown in figure 4.

**4 Application to landslide simulation based on TRIGRS model**

The factor of safety(FS) in TRIGRS is calculated as follows:

$$Fs = \frac{\tan\phi}{\tan\alpha} + \frac{c - \varphi(Z,t)\gamma_W \tan\phi}{\gamma_S Z \sin\alpha\cos\alpha} \tag{20}$$





in which $c$ is soil cohesion, $\alpha$ is slope angle, $\phi$ is soil friction angle, $\varphi$ is the ground-water pressure head depends on depth $Z$ and time $t$, $\gamma_W$ is ground-water unit weight and $\gamma_S$ is soil unit weight. The equation of post-failure motion depending on time is

$$\frac{1}{g}\frac{dv}{dt} = \sin\alpha[1 - Fs(Z,t)]$$

(21)

where $g$ is gravitational acceleration, $v$ is downslope landslide velocity.

An example of 10*10 grid TRIGRS model is set to be the background, and the simulated observations are generated from the $Fs$ by adding a disturbance with normal distribution $N(0.2, 0.3)$. Due to the difficulties of determining the input parameter $\phi$, the soil friction angle, and its highly sensitivity to results, we now generate particle sets of $\phi$, and make $Fs$ be the assimilation variable. The input model variance of $\phi$ is 2 and observation variance of $Fs$ is 0.3. At each step, $\phi$ and $Fs$ will

be updated, and the updated parameters continue to participate in the next step operation as initial parameters. The number of particles is set to 20 in the particle filter program. Figure 5 shows the model results and assimilation results of running for 5 days, 10 days, 15 days, 20 days, respectively.

The root mean square difference of the whole grid of points is calculated to measure the estimated error as follow

$$RMSD_{grid} = \sqrt{\frac{1}{N_p}\sum_{i,j}(\hat{X}_{ij} - X_{ij}^{obs})^2}$$

(22)

where $N_p$ is the total number of grid points, $i, j$ are the indices of rows and columns respectively. The RMSD curve with assimilating days is shown in figure 6 which suggests the value is large in the first 2 days of initialization, fluctuating in next days and steady when there are no observations.

## 5 Conclusion and discussion

The problems of particle degeneration and efficient expression of posterior PDF are long-term difficulties which affect the
performance of particle filter. Many resampling methods can improve effectiveness of particles, but they still need a large number of samples resulting in a large amount of computation.

In this study, we propose two approaches to improve the particle filter process. Firstly, for the problem of particle degeneration, new Gaussian-distributed offspring particles are generated for each mother particle. It can avoid particle duplication and maintain particles' diversity. Secondly, in order to improve the propagating efficiency of a priori particle into a posteriori
particle, an additional item is added which is similar to the Kalman gain at the step of particle propagation, which greatly reduces the number of particles required. It uses only dozens of particles to get good results. Simulating experiment of Lorenz-63 model is carried out to validate the feasibility of these methods. The TRIGRS landslides model is firstly proposed to apply





to the assimilation system. Results show that the assimilating process can make the estimation close to observations, which proved the availability of applying the improved particle filter to landslide model.

However, some disadvantages are still present. Grids are independent of each other in TRIGRS, this leads to the FS estimations are possible to be greater than the actual values. Therefore, the FS estimations only provide references for the actual values. The experiment needs improvement.

**Acknowledgments.** This work is financially supported by the National Key Basic Research Program of China (Grant No. 2013CB733205).





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





## Figures

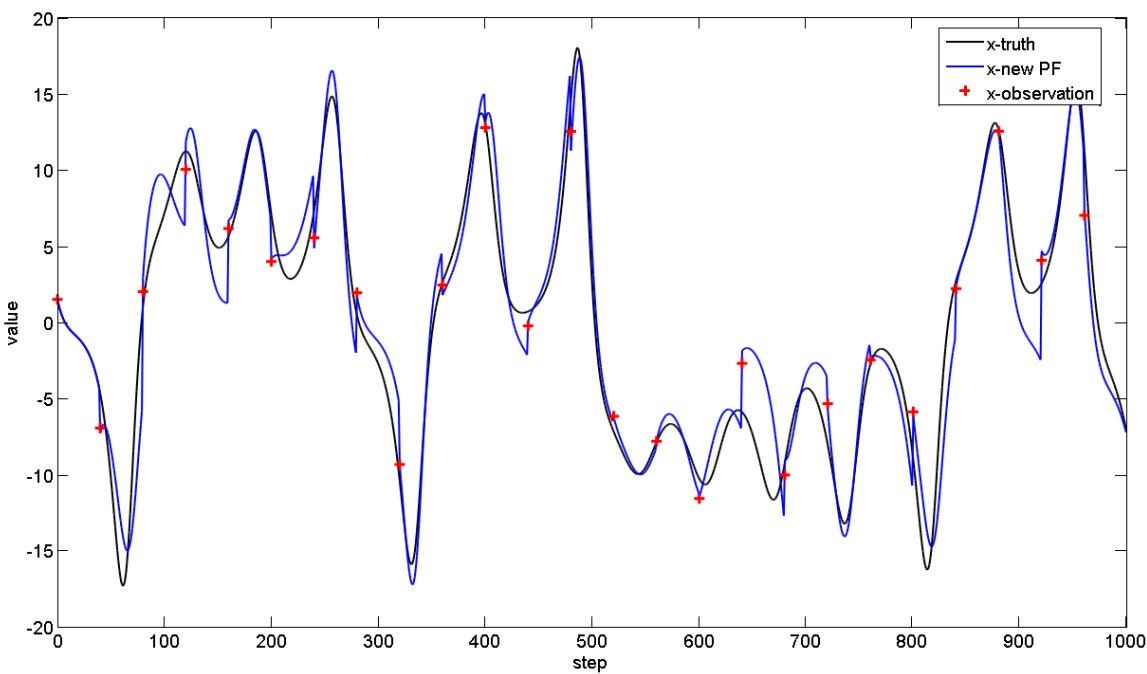

Figure 1: Results of new PF for the Lorenz-63 model of *x*-component. The red crosses are observations, the black line is the true

state and the blue line is the new PF results.



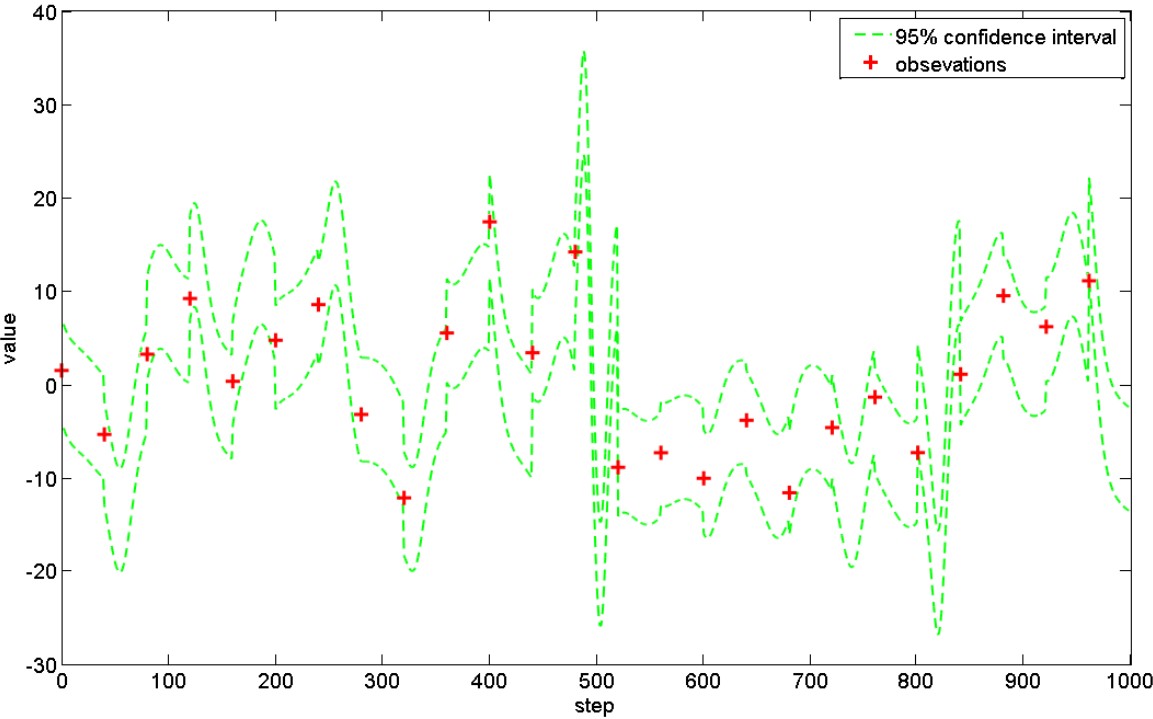

**Figure 2. The 95% confidence interval computed by posterior particles. The green dashed lines donate the upper and lower limits of the interval and the red crosses are observations.**



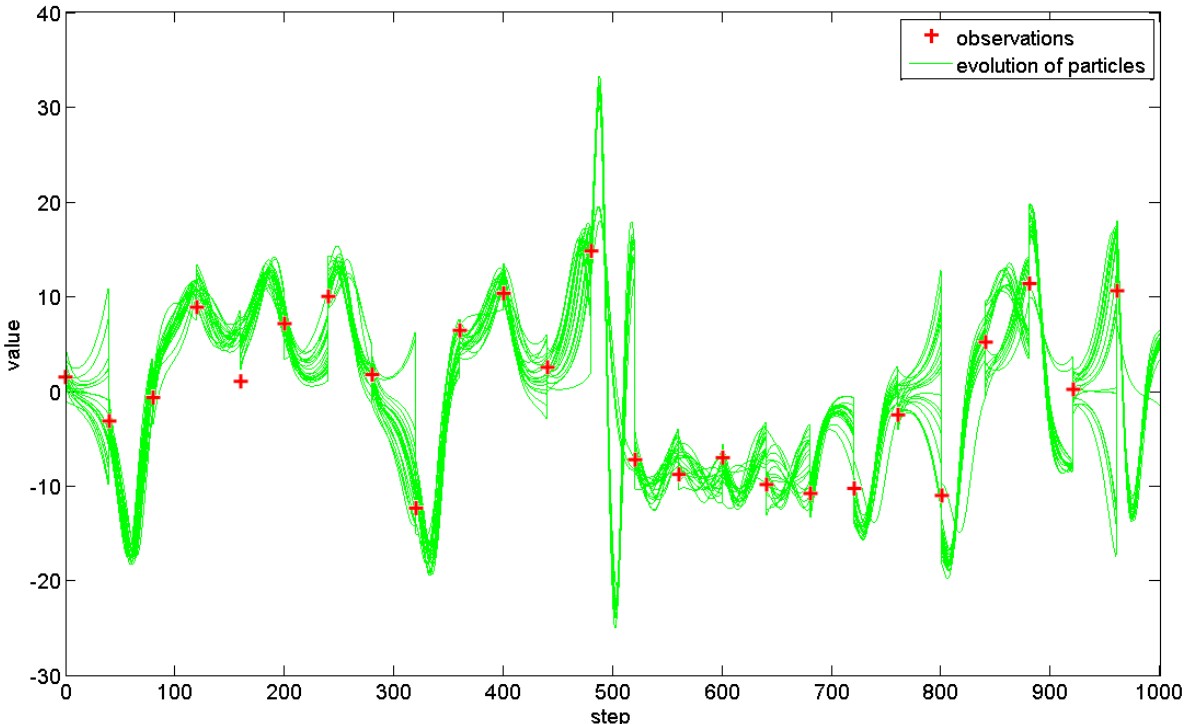

**Figure 3. The evolution of posterior particles in time. The green dashed lines show the traces of all particles, the red crosses donate the observations.**





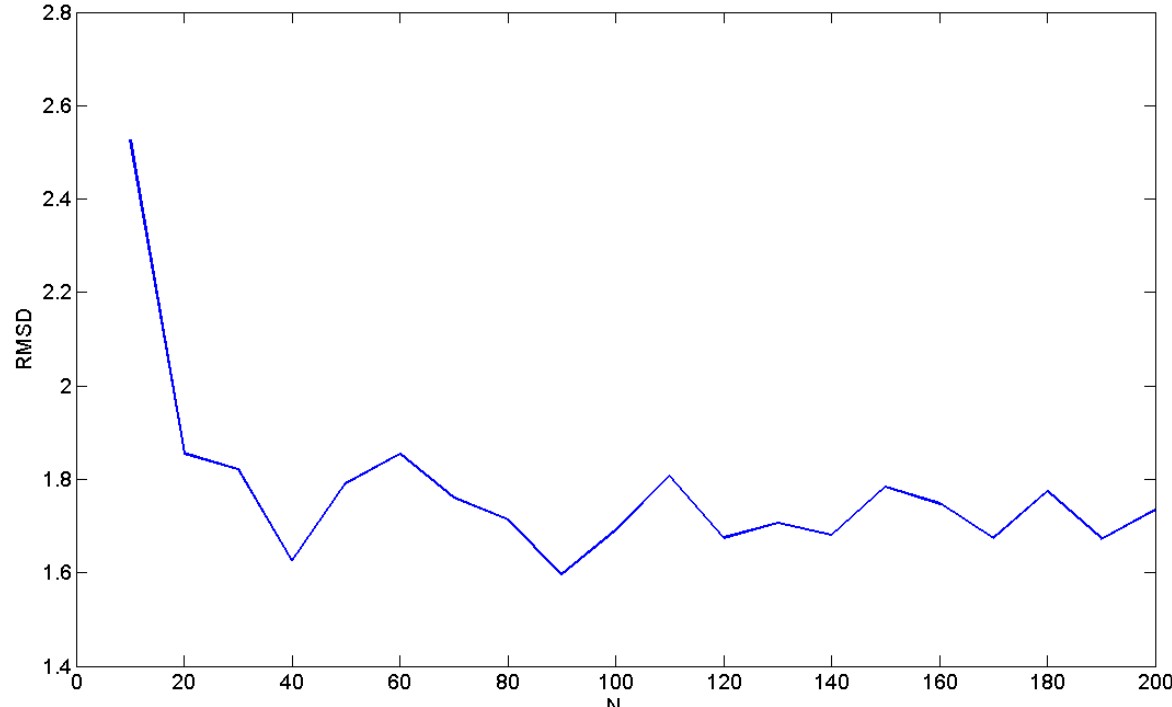

**Figure 4. RMSD of the estimation with respect to particle numbers.**





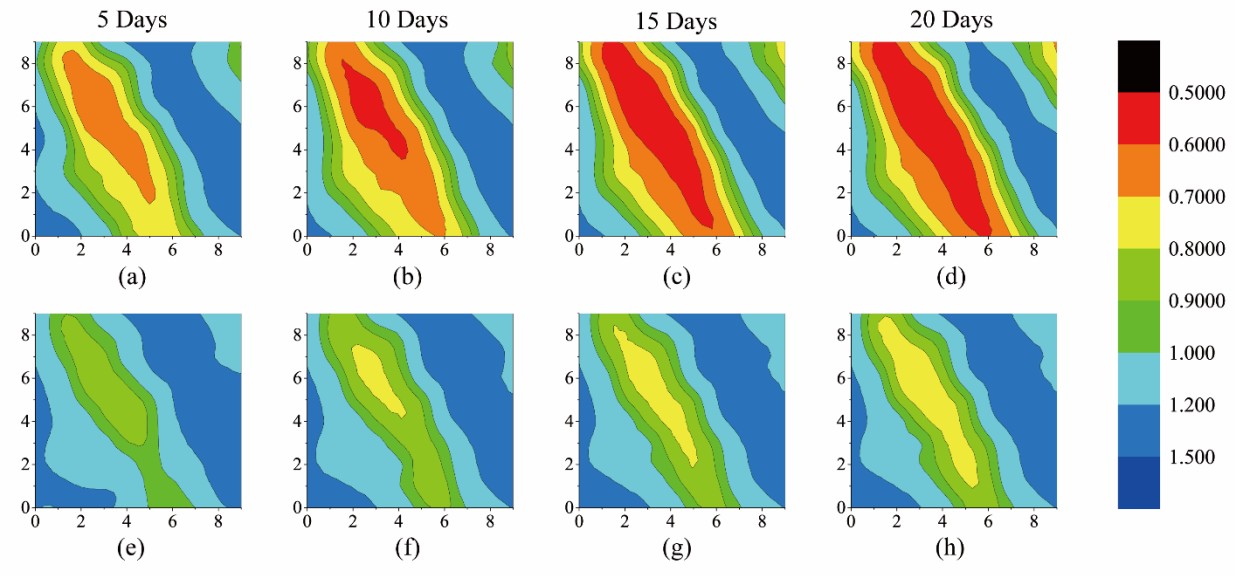

**Figure 5. Model results and assimilation results. The maps in the first row are the model running for 5, 10, 20 days respectively, and that in the second row are the assimilation results.**



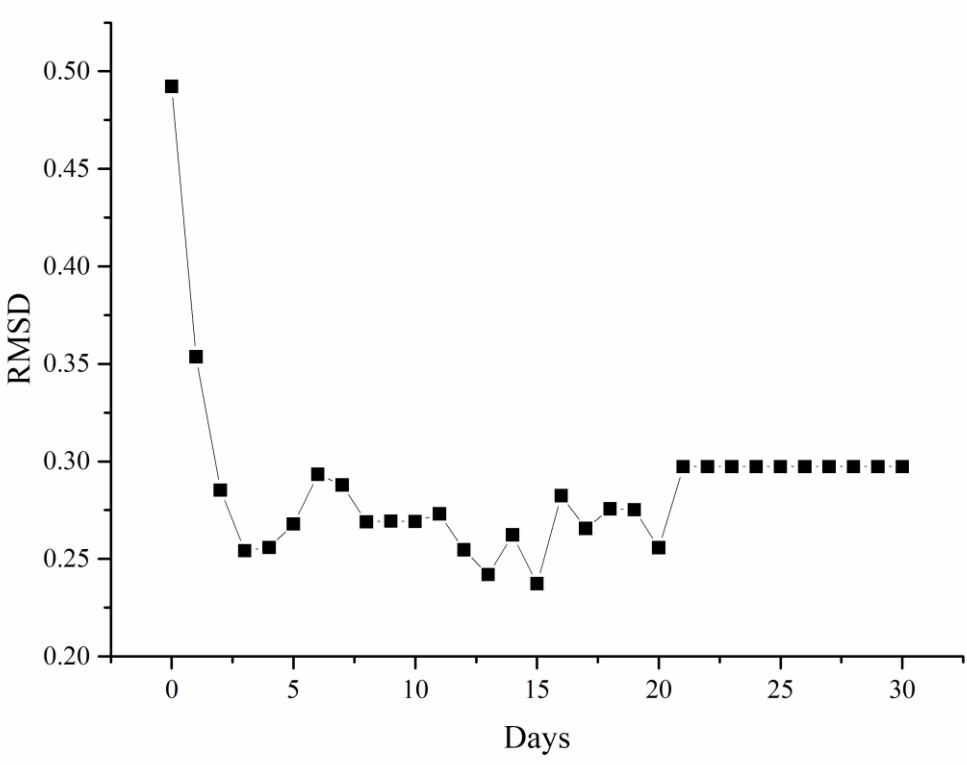

**Figure 6. RMSD line of all grids depending on assimilating time. The TRIGRS model is assimilated with observations in the first 20 days, and results of 21~30th days are model-running results without observations assimilated.**