# Peer review of "Data Assimilation with an Improved Particle Filter and its"

_Natural Hazards and Earth System Sciences, 2017_

## Referee Comment (RC1) · Anonymous Referee #1 · 6 Mar 2018

General comments

The paper presents the results of a study aimed at improving a data assimilation (DA) algorithm based on the residual resampling particle filtering. Two applications are provided in order to respectively test the feasibility of the improved algorithm and show an example concerning slope instabilities. In this latter regard, a 'synthetic' case is presented starting from the expression of the factor of safety implemented in the TRIGRS physically-based model. From this point of view, in the abstract the positive effects of the proposed DA algorithm in the use of TRIGRS should be enhanced and, more in general, the main goal to be pursued with reference to slope stability processes should be more clearly stated. Indeed, the submitted version of the paper does not allow understanding the benefits deriving from the adoption of the improved algorithm in ad-

dressing practical issues about landslides. In my opinion, for the readers of NHESS International Journal, the paper could be of interest only if the theoretical approach is applied to a real (not to a synthetic) case study. Finally, the paper is poorly written and, in some parts, difficult to understand; in this regard, the manuscript needs some English language editing.

Specific comments

Introduction – page 1, line 17. Why the (only) landslide event occurred in China on June 24, 2017 is mentioned? Section 1, Introduction – page 1, lines from 19 to 23. Considering the scope of the paper, why the authors mentioned some numerical methods for landslide modelling? And what type of landslides the authors are taking into account? The description of the TRIGRS model is very poor and should be improved. Section 1, Introduction – page 2, lines from 20 to 22. As mentioned in the general comments, the manuscript includes some sentences that appear meaningless. For example, the authors claim that they choose a 'slope movement model' (?) with a 10*10 size grid (no information about the dimensions are provided), applying the assimilation algorithm and TRIGRS program to 'predict and improve the prediction' (?) of safety factors (more than one?) and deformations (TRIGRS does not allow studying deformations) of the landslide (which?). Section 4, Application to landslide simulation based on TRIGRS model – page 6, lines from 1 to 5. Bearing in mind that TRIGRS allows simulating only the triggering stage of landslides, why the authors considered the post-failure stage? And, once again, what type of rainfall-induced landslide are they referring to? Or, more in general, what kind of physical process are they simulating and how the variation with time of the groundwater pressure head is estimated? Section 4, Application to landslide simulation based on TRIGRS model – page 6, lines from 11 to 12. Could the authors clarify the meaning of Figure 5? Numbers in Figure are representative of what? And colour shadings?

Technical corrections

Section 1, Introduction – page 1, lines 21 and 22. Iverson did not carry out the TRIGRS program; as a matter of fact, the TRIGRS model performs transient seepage analyses using the linearised solution of Richards' equation proposed by Iverson (2000). Please correct accordingly. Section 1, Introduction – page 1, line 23. Please modify "Baum 2008" with "Baum et al. 2008". Section 1, Introduction – page 1, line 23. "Jiang" is not included in the references. Section 1, Introduction – page 2, line 17. The acronym "PDF" is introduced without explanation. Section 3, Application to Lorenz-63 model – page 5, line 5. Greek symbols are introduced without explanation. Section 3, Application to Lorenz-63 model – page 5, line 7. Please modify "are generated form the" with "are generated from the". Section 3, Application to Lorenz-63 model – page 5, lines 10, 13 and 14. "closed to" or "close to"? Section 3, Application to Lorenz-63 model – page 5, line 19. Please modify "RMSE" with "RMSD". Section 4, Application to landslide simulation based on TRIGRS model – page 6, line 1. Please modify "depends on" with "depending on". Section 4, Application to landslide simulation based on TRIGRS model – page 6, line 2. Please modify "is the soil unit weight" with "is the soil unit weight at saturation". Section 5, Conclusion and discussion – page 7, line 3. Please modify "Grids are independent of" with "Grid cells are independent of".

---

## Referee Comment (RC2) · Anonymous Referee #2 · 11 Mar 2018

This paper proposes an improved particle filter to enhance the performance of data assimilation in TRIGRS landslide model. The study fit the topic of the journal. This paper can be published while following questions are clarified.

1. Extensive editing of English language and style required: this must be reviewed in depth.

2. The improvements such as the accuracy and computation burden of the particle filter should be more clarified

3. Section 4, the authors mentioned "observations are generated from the Fs by adding a disturbance with normal distribution N(0.2, 0.3)", why the mean of disturbances is 0.2 rather than 0?

[Figure]

4. I noticed that the FS was chosen as the assimilated factor, why not use the displacement?

5. Data assimilation is usually applied on large scale scenarios. This study employed assimilation size 10*10, I suggest you increase the assimilation size, or use true landslide monitoring data instead.

Comments: 1. The full name of "TRIGRS" should be given at its first appearance.

2. Page 1 Line 7 and 8, I think it would be better to recognize this sentence.

3. Page 1 Line 23, reference missing: 'Jiang adopted the Ensemble Kalman filter to landslide movement model in relation to hydrological factors, which introduce data assimilation (DA) to landslide.'

4. Page 2 Line 14: 'It can get good results to using. . ...' should be 'to use'

---

## Author Comment (AC1) · 24 Apr 2018

Thanks for your comments. The following is my reply.

General comments: The paper presents the results of a study aimed at improving a data assimilation (DA) algorithm based on the residual resampling particle filtering. Two applications are provided in order to respectively test the feasibility of the improved algorithm and show an example concerning slope instabilities. In this latter regard, a 'synthetic' case is presented starting from the expression of the factor of safety implemented in the TRIGRS physically-based model. From this point of view, in the abstract the positive effects of the proposed DA algorithm in the use of TRIGRS should be enhanced and, more in general, the main goal to be pursued with reference to slope

stability processes should be more clearly stated. Indeed, the submitted version of the paper does not allow understanding the benefits deriving from the adoption of the improved algorithm in addressing practical issues about landslides. In my opinion, for the readers of NHESS International Journal, the paper could be of interest only if the theoretical approach is applied to a real (not to a synthetic) case study. Finally, the paper is poorly written and, in some parts, difficult to understand; in this regard, the manuscript needs some English language editing. Reply: In this paper, a synthetic experiment is presented to verify the feasibility of the algorithm and its application to the TRIGRS landslide model. The main goal of this study is to propose a new method and prove it can be applied to the evaluation of FS in landslide slope. Experiments of real cases is carrying out and it need some more monitoring data. Then the paper is modified in some poorly expressed places to improve the expression of English languages.

Specific comments: Introduction – page 1, line 17. Why the (only) landslide event occurred in China on June 24, 2017 is mentioned? Section 1, Introduction – page 1, lines from 19 to 23. Considering the scope of the paper, why the authors mentioned some numerical methods for landslide modelling? And what type of landslides the authors are taking into account? The description of the TRIGRS model is very poor and should be improved. Section 1, Introduction – page 2, lines from 20 to 22. As mentioned in the general comments, the manuscript includes some sentences that appear meaningless. For example, the authors claim that they choose a 'slope movement model' (?) with a 10*10 size grid (no information about the dimensions are provided), applying the assimilation algorithm and TRIGRS program to 'predict and improve the prediction' (?) of safety factors (more than one?) and deformations (TRIGRS does not allow studying deformations) of the landslide (which?). Section 4, Application to landslide simulation based on TRIGRS model – page 6, lines from 1 to 5. Bearing in mind that TRIGRS allows simulating only the triggering stage of landslides, why the authors considered the post-failure stage? And, once again, what type of rainfall-induced landslide are they referring to? Or, more in general, what kind of physical process are they simulating and how the variation with time of the groundwater pressure head is estimated? Section 4,

Application to landslide simulation based on TRIGRS model – page 6, lines from 11 to 12. Could the authors clarify the meaning of Figure 5? Numbers in Figure are representative of what? And color shadings? Reply: Some extra content has been deleted, such as the landslide event occurred in China on June 24, 2017. In section 1, some methods for landslide modeling are mentioned to introduce the research status of landslide deformation analysis and numerical landslide evaluation. This study is applied to "peristaltic landslides", which is added in the last paragraph of section 1. The description of the TRIGRS model is enriched in the beginning of section 4. In the manuscript, poor expressed contents mentioned in the comment have been modified. In section 4, the useless content of post failure stage has been deleted. To estimate the groundwater pressure head ($\varphi$), some content of $\varphi$-estimation is added to the manuscript. Formula (21) and its context is the calculation method of $\varphi$-estimation, and Figure 6 and Figure 7 are its change of overall distribution and single cell, respectively. The illustration of Figure 5 has been revised to "Model results and assimilation results of FS. The maps in the first row are the model results running for 5, 10, 15, 20 days respectively, and that in the second row are the assimilation results. The horizontal and vertical coordinates in each graph are grid numbers of each cell."

Technical corrections Thanks for your review. The manuscript has been revised.

Please also note the supplement to this comment:
https://www.nat-hazards-earth-syst-sci-discuss.net/nhess-2017-439/nhess-2017-439-AC1-supplement.zip
* * *
[Figure]

5 Days  10 Days  15 Days  20 Days  FS

(a)  (b)  (c)  (d)

(e)  (f)  (g)  (h)

Figure 5. Model results and assimilation results of FS. The maps in the first row are the model results running for 5, 10, 15, 20 days respectively, and that in the second row are the assimilation results. The horizontal and vertical coordinates in each graph are grid numbers of each cell.

**Fig. 1.**

[Figure]

Figure 6. The distribution variation of groundwater pressure head ( $\varphi$ ) with assimilated time. The horizontal and vertical coordinates in each graph are grid numbers of each cell.

15

**Fig. 2.**

Figure 7. The changing line of the groundwater pressure head ($\varphi$) estimation of grid cell (5, 5) with assimilating time. The value is

growing with the evolution of the landslide.

16

**Fig. 3.**

---

## Author Comment (AC3) · 24 Apr 2018

Thanks for your comments. The following is my reply.

Questions reply: 1. Extensive editing of English language and style required: this must be reviewed in depth. Reply: The manuscript has been revised. The text is modified in some poorly expressed places to improve the expression of English languages. 2. The improvements such as the accuracy and computation burden of the particle filter should be more clarified. Reply: At the end of section 2, the root mean square difference (RMSD) has been added as a measure factor to evaluate the accuracy. The main computation burden of the particle filter is explained in Para.2 of Sec.2: "Residual resample is a way to solve the problem of particle degeneracy which is an unavoidable

trouble in standard PF. With the recursive progress, the weights of particles are gradually concentrated on a few samples and others tend to be zero. To keep most particles effective, low-weight particles are removed and high-weight particles are duplicated. This causes that the particle sets can hardly represent the prior PDF due to the declining of particles diversity." 3. Section 4, the authors mentioned "observations are generated from the Fs by adding a disturbance with normal distribution N(0.2, 0.3)", why the mean of disturbances is 0.2 rather than 0? Reply: Due to the TRIGRS model calculate the safe factor cell by cell, without considering the interaction force between grid cells, the TRIGRS output results have systematic errors. So, we assumed a disturbance with an experience mean of 0.2. Additionally, the estimation of parameter $\varphi$ has been increased in section 4. Figure 6 and Figure 7 are distribution and change line of $\varphi$ respectively. 4. I noticed that the FS was chosen as the assimilated factor, why not use the displacement? Reply: In the post failure stage of landslide, the two variables, FS and displacement (in fact the integration of displacement velocity over time, dv/dt), can be converted to each other. The FS determines the integration of displacement velocity over time. When the displacement is chosen as the assimilated factor, it is necessary to convert the FS to velocity, and then accumulate to get displacement by time. This progress would magnify the error of FS, and the difference between model value of displacement and the observation would be larger. That would reduce the efficiency of particle filter. To convert the displacement to FS can control the dispersion of errors. Besides, this also reduces computational complexity. Therefore, FS is more suitable to be the assimilated factor than displacement. 5. Data assimilation is usually applied on large scale scenarios. This study employed assimilation size 10*10, I suggest you increase the assimilation size, or use true landslide monitoring data instead. Reply: In the 3rd paragraph of section 4, the size of the assimilation area has been increased. "An example of 10 * 10 grid TRIGRS model is set to be the background, and each grid cell is a square with a length of 10 meters." In this paper, a synthetic experiment is presented to verify the feasibility of the algorithm and its application to the TRIGRS landslide model. The main goal of this study is to propose a new method and prove it

can be applied to the evaluation of FS in landslide slope. Experiments of real cases is carrying out and it need some more monitoring data.

Comments reply: 1. The full name of "TRIGRS" should be given at its first appearance. Reply: Thanks. The full name of "TRIGRS" is added in the first paragraph of introduction. 2. Page 1 Line 7 and 8, I think it would be better to recognize this sentence. Reply: The manuscript has been modified to "In this work, an improved particle filter algorithm is proposed. To overcome the particle degeneration and improve particles' efficiency, the processes of particle resample and particle transferring are updated." 3. Page 1 Line 23, reference missing: 'Jiang adopted the Ensemble Kalman filter to landslide movement model in relation to hydrological factors, which introduce data assimilation (DA) to landslide.' Reply: The reference has been added. "Jiang, Y. A., M. S. Liao, Z. W. Zhou, X. G. Shi, L. Zhang and T. Balz (2016). "Landslide Deformation Analysis by Coupling Deformation Time Series from SAR Data with Hydrological Factors through Data Assimilation." Remote Sensing 8(3)." 4. Page 2 Line 14: 'It can get good results to using...' should be 'to use'. Reply: Thanks. The manuscript has been modified. Some other expression errors have also been modified. The supplement is the modified manuscript.

Please also note the supplement to this comment:
https://www.nat-hazards-earth-syst-sci-discuss.net/nhess-2017-439/nhess-2017-439-AC3-supplement.pdf

———————————————

[Figure]

[Figure]

Figure 6. The distribution variation of groundwater pressure head ($\varphi$) with assimilated time. The horizontal and vertical coordinates in each graph are grid numbers of each cell.

15

**Fig. 1.**

Figure 7. The changing line of the groundwater pressure head ( $\varphi$ ) estimation of grid cell (5, 5) with assimilating time. The value is

growing with the evolution of the landslide.

16

**Fig. 2.**